# Fusion guide RNAs for orthogonal gene manipulation with Cas9 and Cpf1

Jiyeon Kweon[1], An-Hee Jang[1], Da-eun Kim[2,3], Jin Wook Yang[1], Mijung Yoon[1], Ha Rim Shin[1], Jin-Soo Kim[2,3] & Yongsub Kim [1]

The bacteria-derived clustered regularly interspaced short palindromic repeat (CRISPR)–Cas systems are powerful tools for genome engineering. Recently, in addition to Cas protein engineering, the improvement of guide RNAs are also performed, contributing to broadening the research area of CRISPR–Cas9 systems. Here we develop a fusion guide RNA (fgRNA) that functions with both Cas9 and Cpf1 proteins to induce mutations in human cells. Furthermore, we demonstrate that fgRNAs can be used in multiplex genome editing and orthogonal genome manipulation with two types of Cas proteins. Our results show that fgRNAs can be used as a tool for performing multiple gene manipulations.

[1] Department of Biomedical Sciences, University of Ulsan College of Medicine, Asan Medical Center, Seoul, 05505, Republic of Korea. [2] Department of Chemistry, Seoul National University, Seoul, 08826, Republic of Korea. [3] Center for Genome Engineering, Institute for Basic Science, Seoul, 08826, Republic of Korea. Correspondence and requests for materials should be addressed to Y.K. (email: yongsub1.kim@gmail.com)

Clustered, regularly interspaced, short palindromic repeat (CRISPR)-Cas9 systems are widely used in the field of genome manipulation for gene editing and transcriptional and epigenetic perturbation[1]. Recently, others and we have shown that another Cas protein, Cpf1 (class 2, type V), can be successfully repurposed for genome editing of eukaryotic cells, animals, and plants[2–6]. Although both Cas9 and Cpf1 proteins are guide RNA (gRNA)-mediated endonucleases, the Cpf1 protein has several distinct properties compared with Cas9 protein (e.g., T-rich protospacer adjacent motif (PAM) sequences, PAM-distal region cutting, 5′-overhang sticky-end DSBs, and RNase activity), which provide significant increase in the potential application of the CRISPR-based genome manipulation toolbox.

In addition to the considerable effort expended on improving and developing Cas proteins[7–12], gRNA engineering is being performed to refine the CRISPR–Cas9 and Cpf1 systems. Additional 5′-guanine or truncated gRNAs can reduce the mismatch tolerance and increase target specificity without sacrificing on-target activity[13,14]. Moreover, in addition to improving the specificity of CRISPR–Cas9 systems, gRNA engineering has broadened the set of potential applications of CRISPR–Cas9 systems. For example, aptamer-fused Cas9 gRNAs can recruit additional effector proteins, such as VP64, KRAB, and APOBEC, for genome manipulation[15–17]. Moreover, Cpf1 gRNAs have been engineered to simplify multiplex genome editing by exploiting the

crRNA processing activity of the Cpf1 protein[18]. In this study, we develop a gRNA engineering method by creating a synthetic fusion gRNA (fgRNA) that can interact with both Cas9 and Cpf1 proteins. Using the fgRNAs, we demonstrate that fgRNAs can simultaneously induce endogenous mutations at both target sites of Cas9 and Cpf1, and that the mismatch tolerance of the fgRNAs is similar to that of conventional gRNAs. Furthermore, we show that the fgRNAs can work with dCas9 variants and Cpf1 to induce gene activation and disruption at endogenous target sites. On the basis of these results, fgRNAs have the potential to expand multiple gene manipulation using the orthogonality of various types of Cas proteins.

## Results

**fgRNAs mediated endogenous mutation.** First, we carefully compared the gRNA composition of the Cas9 and Cpf1 systems. In CRISPR–Cas9 systems, the chimeric single gRNA (sgRNA) requires a tracrRNA fused at the 3′-end of the crRNA. In CRISPR–Cpf1 systems, crRNAs are composed of target-specific gRNA with a 5′-scaffold. In other words, the 3′-scaffold is necessary for CRISPR–Cas9 gRNAs and the 5′-scaffold is necessary for CRISPR–Cpf1 gRNAs. From these features, we constructed fgRNAs containing both the 5′-scaffold of Cpf1 and the 3′-scaffold of Cas9, with the expectation that the resultant

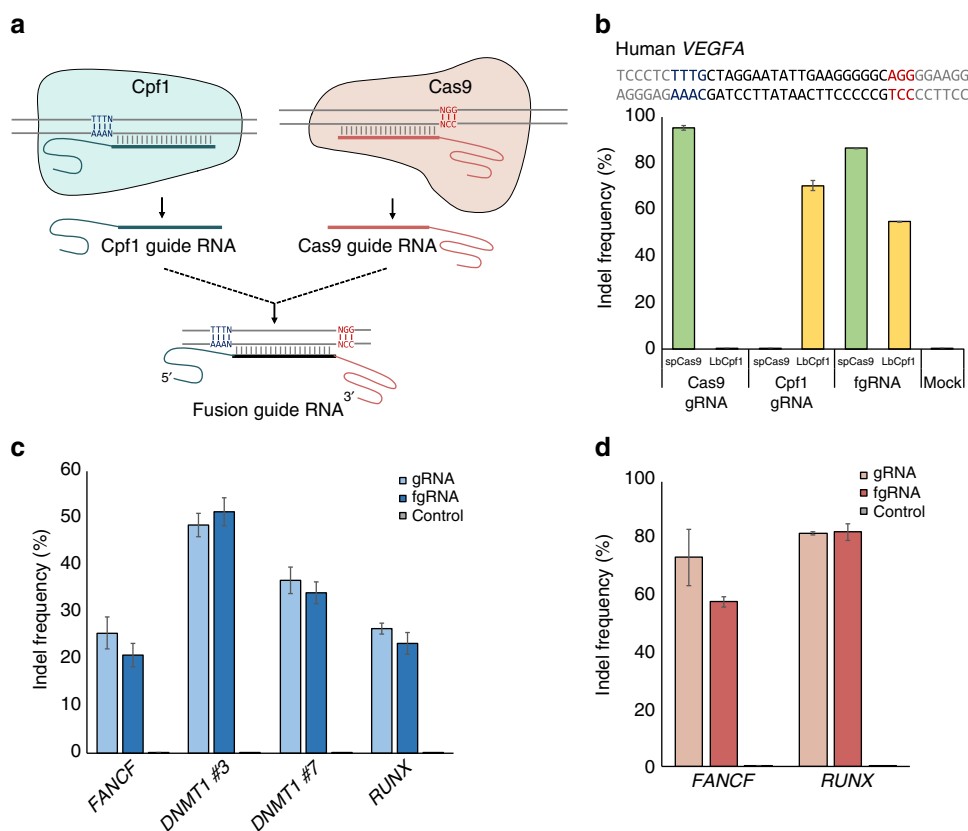

**Fig. 1** fgRNA can induce endogenous mutations. **a** Schematic overview of the fgRNA construct. Cpf1 guide RNAs have 5′-scaffold sequences in front of the target sequences, and Cas9 guide RNAs have 3′-scaffold sequences behind the target sequences. fgRNAs have scaffolds for each Cas protein, enabling genome editing with both Cas9 and Cpf1. **b** Top, human *VEGFA* target sequences. PAM sequences of LbCpf1 and spCas9 are colored in blue and red, respectively. Target sequences of LbCpf1 and spCas9 completely overlap. Bottom, fgRNA-mediated endogenous indel frequencies measured by targeted deep sequencing. spCas9 and LbCpf1 could induce endogenous mutations with both fgRNAs and their guide RNAs. **c, d** Indel frequencies of four additional target sites of LbCpf1 (**c**) and two additional target sites of spCas9 (**d**). Both LbCpf1 and spCas9 can introduce indels using fgRNAs at similar levels to their guide RNAs. Error bars indicate s.e.m. (*n* = 2)

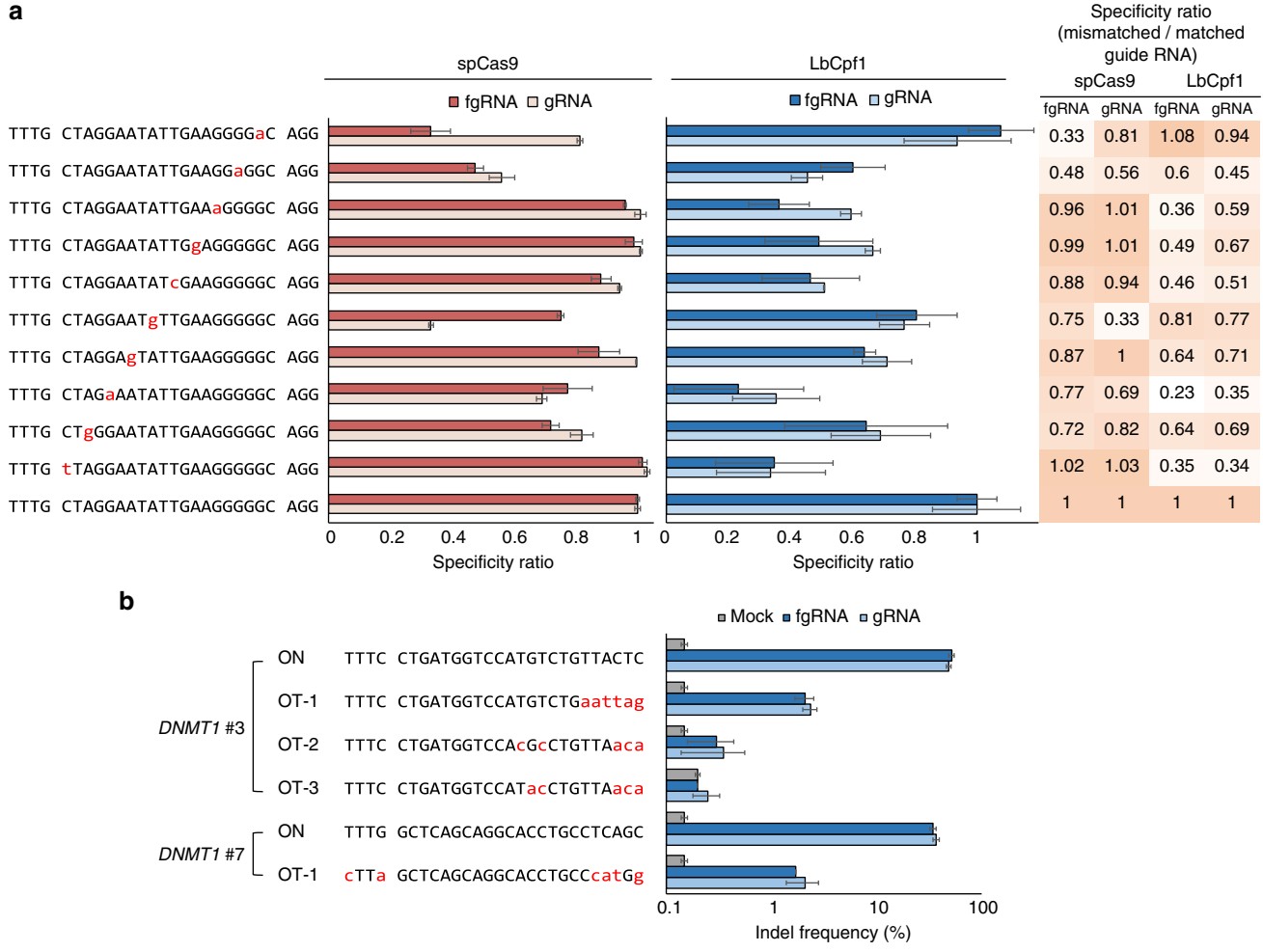

**Fig. 2** Comparison of mismatch tolerance of fgRNAs with those of guide RNAs. **a** Mismatched fgRNAs or gRNAs that differed from human *VEGFA* target sites by one nucleotide were transfected into HeLa cells. **b** Comparison of off-target effects between the fgRNAs and conventional gRNAs of LbCpf1. No significant difference was observed between the two types of guide RNAs at four different potential off-target sites. Mismatched nucleotides are colored in red. Error bars indicate s.e.m. ($n = 2$)

fgRNAs would recruit both Cas9 and Cpf1 proteins to their target sites (Fig. 1a).

To investigate whether fgRNAs can guide both Cas9 and Cpf1 proteins to induce mutations in human cells, we chose *VEGFA* target sites that could be shared by LbCpf1 (*Lachnospiraceae bacterium* ND2006 Cpf1) and spCas9 (*Streptococcus pyogenes* Cas9). Using targeted deep sequencing, we found that fgRNAs could introduce endogenous indels with both Cas9 and Cpf1 proteins with comparable mutation frequencies to those of conventional gRNAs of spCas9 and LbCpf1 (95.5 and 86.9%, respectively, with spCas9 and 70.6 and 55.0%, respectively, with LbCpf1) (Fig. 1b). These findings were consistent with the results of other endogenous target sites, including *FANCF* and *RUNX* (Fig. 1c, d). We also fused fgRNAs with the 5′-scaffold of AsCpf1 (*Acidaminococcus sp.* BV3L6 Cpf1) instead of LbCpf1 and found that fgRNAs could also induce mutations at the endogenous *RUNX* locus in two different human cell lines, namely, HeLa and HEK293T (Supplementary Fig. 1a, b). In contrast to Cas9, Cpf1 possesses both RNase and DNase activities, which allow it to process its own pre-crRNA and subsequently use the processed RNA to recognize and cleave the target DNA. We also confirmed that fgRNAs recruit both Cpf1 and Cas9 proteins without any modification and might function with both Cpf1 and Cas9 proteins at each targeted locus (Supplementary Fig. 2). These

results showed that fgRNAs can be used with both Cas9 and Cpf1 in targeted mutagenesis in human cells.

**Potential off-target effects of fgRNAs**. To assess the specificity of fgRNAs, we constructed a series of mismatched fgRNAs targeting human *VEGFA* sites and analyzed their mutagenesis activities using targeted deep sequencing (Fig. 2a). We found that fgRNAs had a similar level of mismatch tolerance to the conventional gRNAs of LbCpf1 and spCas9. We also investigated the specificity of human *DNMT1*-targeting fgRNA at several endogenous potential off-target sites defined in previous studies[19,20]. Notably, fgRNAs exhibited indel frequencies comparable to those of Cpf1 gRNAs at all potential off-target sites in HeLa cells (Fig. 2b).

**Multiplex genome editing with fgRNAs**. We next tested whether fgRNAs could be used in multiplex genome editing with Cas9 and Cpf1 proteins. For this purpose, the length of target-specific sequences between the 5′- and 3′-scaffolds were extended up to 40-bp for two different target sites. In preliminary experiments, we compared the mutagenesis activities of five fgRNAs containing sequences of different lengths between each scaffold and observed whether these fgRNAs could induce indels at human endogenous *VEGFA* sites. With both spCas9 and LbCpf1 proteins, no

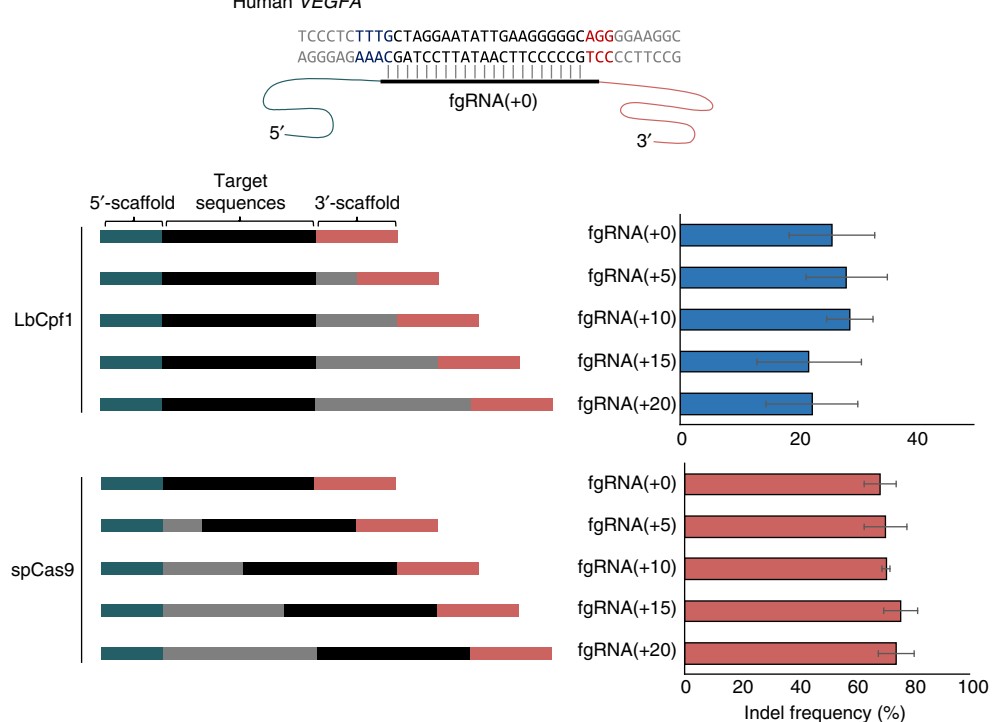

**Fig. 3** Endogenous indel frequencies according to the lengths of the target sequences between the 5′- and 3′-scaffolds. Top, human *VEGFA* target sequences and fgRNAs are shown. PAM sequences of LbCpf1 (TTTG) and spCas9 (AGG) are colored blue and red, respectively. The mimetic construct of fgRNAs is described; blue–green and pink colors represent 5′-scaffolds of LbCpf1 and 3′-scaffolds of spCas9, respectively. Bottom, endogenous indel frequencies are shown based on different lengths of fgRNAs, as measured using targeted deep sequencing. Each target sequence is listed in Supplementary Table 1. Error bars indicate s.e.m. ($n = 2$)

significant difference was observed in mutagenesis activities among all lengths of fgRNAs (22.5–28.8% with LbCpf1 and 68.6–75.8% with spCas9; Fig. 3). On the basis of these results, we hypothesized that fgRNAs could be simultaneously used to target two different genes with LbCpf1 and spCas9 (Fig. 4a). By co-transfecting fgRNA with LbCpf1 and spCas9, we successfully induced indels at endogenous *VEGFA* and *FANCF* loci (24.3 and 57.3%, respectively) and at *VEGFA* and *TREX2* loci (29.9 and 64.1%, respectively) (Fig. 4b, c). We also successfully induced ssODN-mediated HDR in the *VEGFA* locus and gene disruption of the *PRKDC* locus in the same cells using fgRNAs (Supplementary Fig. 3).

**Orthogonal genome manipulation with fgRNAs.** Finally, we investigated whether fgRNAs could be used for orthogonal genome manipulation, especially for gene knockout and transcriptional activation of two different genes (Fig. 4a). For this purpose, we constructed fgRNAs containing target sequences of both endogenous *MYOD* and *VEGFA* genes. *MYOD* genes were successfully activated using a modified CRISPR–Cas9 system in a previous study[21], and we designed fgRNAs that could simultaneously activate the *MYOD* gene and disrupt the *VEGFA* gene. By co-transfecting fgRNA-encoding plasmids with a MS2 loop, LbCpf1, dCas9-VP64, and MCP-p65-HSF1, we observed that *MYOD* mRNA expression levels increased >80-fold and found indel frequencies at the *VEGFA* locus to be 29.0% (Fig. 4d, e). We also observed orthogonal gene activation and disruption of endogenous *ASCL1* and *DNMT1* genes (Supplementary Fig. 4). These results revealed that fgRNAs could indeed be used in orthogonal gene disruption and activation with Cas9 variants and Cpf1 proteins. Taken together, these data highlight the utility of

fgRNAs that recruit both Cas9 and Cpf1 proteins for efficient genome editing in human cells.

## Discussion

CRISPR–Cas systems are widely used in various biological research fields for genome manipulation, and several efforts have been made to improve the nature of these systems. For example, Cas9 and Cpf1, which are representative proteins of the CRISPR system, are engineered for altering PAM sequences and improving target specificities[8,9,14,22]. Also, catalytically inactive Cas9 and Cpf1 are developed and widely used as programmable DNA-binding proteins for transcriptional and epigenetic perturbations[1]. As well as Cas proteins engineering, their gRNAs are also improved to increase indel frequencies, target specificities, and induce multiple genome editing[13,14,18,23].

Cas9 and Cpf1 proteins have distinct characteristics, among which they have different gRNA structures. For this reason, it is necessary to make independent gRNAs even if they recognize the same target sequences. This is why there are no attempt to use the orthogonality of these two different types of Cas proteins, even though they might be complementary because of their distinct characteristics. In this study, we developed and validated fgRNAs —new gRNA constructs that can work with both Cas9 and Cpf1 proteins. In addition, by extending the target sequences of fgRNAs, we demonstrated that fgRNAs can be used for targeting two independent endogenous sites using Cas9 and Cpf1 proteins. We also demonstrated that fgRNAs could be used to induce gene disruption and activation simultaneously. With this concept, we believe that a variety of applications will be possible. For example, if fgRNAs are designed to direct dCas9 variants to key factors of DNA repair mechanisms and Cpf1 to wanted-target genes, it may

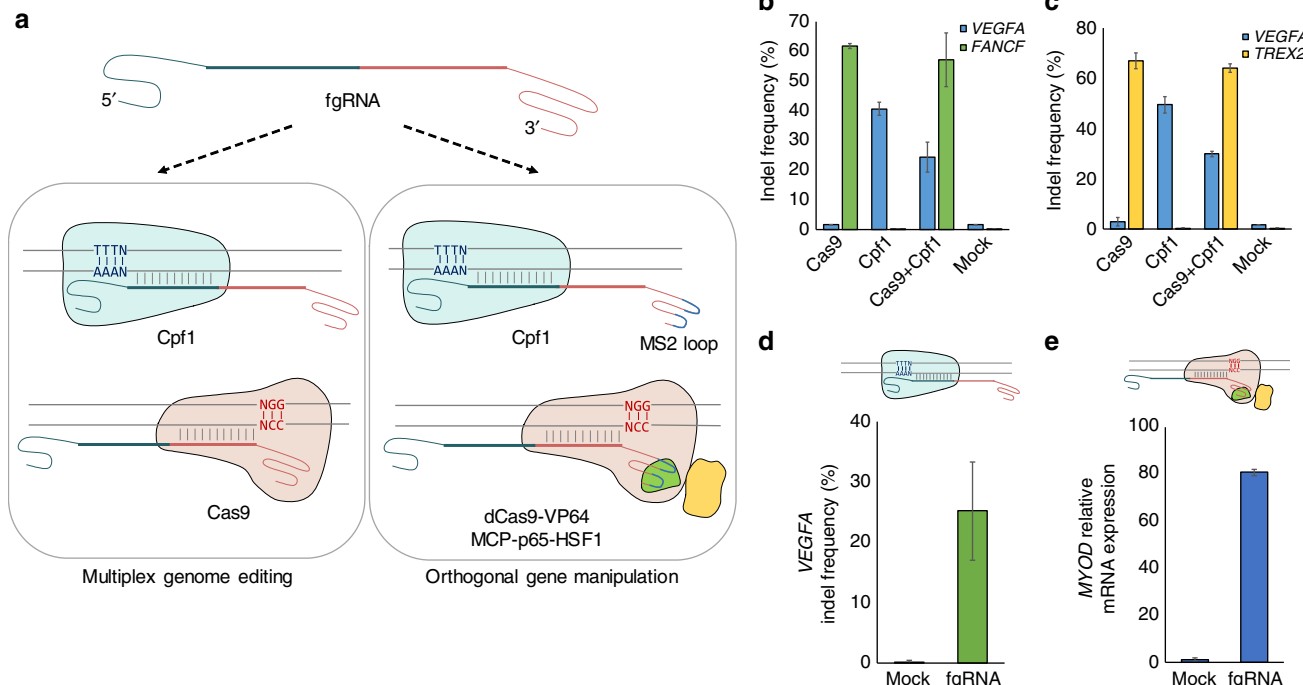

**Fig. 4** Multiplex genome manipulations with fgRNAs. **a** Schematic overviews of multiplex genome editing and orthogonal genome manipulation. fgRNAs containing two different target sequences of Cpf1 and Cas9 could be used in multiplex genome editing with Cpf1 and Cas9. If these fgRNAs are used with Cpf1 and the dCas9 effector (i.e., dCas9-VP64), genome editing and regulation can be simultaneously performed. **b**, **c** Multiple genome editing with three different fgRNAs. fgRNAs were used with Cpf1 and Cas9 to target human *VEGFA* and *FANCF* (**b**), and human *VAGFA* and *TREX2* (**c**). **d**, **e** Orthogonal gene disruption at the endogenous *VEGFA* locus (**d**) and transcriptional activation of endogenous *MYOD* genes (**e**). Error bars indicate s.e.m. ($n = 2$ or 3)

be possible to alter the mutation frequencies or patterns of Cpf1. In addition to the fgRNA itself, fgRNAs combined with the above-mentioned engineered-Cas proteins and improved-gRNAs technologies would give rise to various possibilities applications. Taken together, fgRNAs have the potential to simplify multiple gene manipulation with various Cas proteins and expand the use of genome editing tools in biological research.

## Methods

**Construction of plasmid DNA**. We used human codon-optimized spCas9 [p3s-Cas9HC (Addgene plasmids #43945)], LbCpf1 [pY016 (Addgene plasmids #69988)], and AsCpf1 [pY010 (Addgene plasmids #69982)]. Plasmid DNAs for orthogonal gene manipulation, lenti dCAS-VP64_Blast (Addgene plasmid #61425) and lenti MS2-p65-HSF1_Hygro (Addgene plasmid #61426) were gifts from Feng Zhang. pU6-As-crRNA (Addgene plasmid #78956) and pU6-Lb-crRNA (Addgene plasmid #78957) were used for cloning the gRNA of LbCpf1 and AsCpf1. To construct fgRNA-cloning vector, 5′-scaffold sequences of LbCpf1 (5′-AATTTC-TACTAAGTGTAGAT-3′) and AsCpf1 (5′-TAATTTCTACTCTTGTAGAT-3′) were inserted behind the U6 promoter sequences of the Cas9 gRNA-cloning vector. The target sequences of each gRNA are listed in Supplementary Table 1.

**Cell culture and transfection**. HEK293T/17 (ATCC, CRL-11268) and HeLa (ATCC, CCL-2) cell lines were maintained in DMEM (WelGENE Inc.) supplemented with 10% FBS. 24 h before transfection, $2 \times 10^5$ HEK293T cells and $8 \times 10^4$ HeLa cells were seeded in 24-well plates (Corning). All transfection experiments were conducted using Lipofectamine 2000 (Life Technologies) according to the manufacturer's protocol. Cas protein expression plasmid DNA and gRNA-encoding plasmid DNA were transfected in each well of the 24-well plates in a 1:1 ratio. Each transfection was performed in duplicate. For ssODN-mediated HDR, $2 \times 10^5$ HeLa cells were nucleofected with spCas9 expression plasmid (200 ng), LbCpf1 expression plasmid (200 ng), *VEGFA–PRKDC*-targeting fgRNA (400 ng) and ssODN (100 pmol) according to the manufacturer's protocols (Lonza). To perform single-clone analysis, cells were seeded in a 96-well plate at 0.5-cells per well density and single clones were analyzed 1 week after seeding. Cells were not tested for mycoplasma contamination.

**NGS analysis for measuring mutation frequencies**. Genomic DNA was extracted 72 h after transfection using the DNeasy Blood & Tissue Kit (Qiagen) according to the manufacturer's protocol. The target region was amplified using Phusion High-Fidelity DNA Polymerase (New England Biolabs), and the PCR amplicons were subjected to paired-end sequencing using the Illumina MiSeq. The PCR primer sequences are listed in Supplementary Table 2. We used Cas-Analyzer (http://www.rgenome.net/cas-analyzer/) to analyze the mutation frequencies.

**RNA expression level analysis**. To analyze expression levels of *MYOD* and *ASCL1* mRNA, total RNA was extracted using Riboclear (GeneAll) and cDNA was synthesized using AccuPower® CycleScript RT PreMix (Bioneer). Each cDNA sample was mixed with 2 × SYBR Green SuperMix (Bio-Rad) and subjected to real-time quantitative PCR (qPCR). *MYOD* and *ACDL1* activation levels were calculated using the comparative $C_T$ method. Primer sequences used in qPCR are listed in Supplementary Table 2.

**Data availability**. The data that support the findings of this study are available from the corresponding author upon reasonable request. All targeted deep sequencing data were deposited at NCBI Sequence Reads Archive database with accession number SRP116368.

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

## Acknowledgements
This work was supported by the National Research Foundation of Korea (2016R1D1A1A02937096, 2017M3A9B4062419, and 2016R1A6A3A04009014).

## Author contributions
J.K. and Y.K.: Designed the research. J.K., A.-H.J., D.K., J.W.Y., M.Y., and H.R.S.: Performed the experiments. J.K., J.-S.K. and Y.K.: Provided conceptual advice. J.K. and Y.K.: Wrote the paper. Y.K.: Supervised the research.

## Additional information

**Competing interests:** The authors declare no competing financial interests.

