## [Peer Review File · Nature Communications]

Reviewers' comments:

Reviewer #1:

Remarks to the Author:

In this MS, the authors take advantage of the fact that Cpf1 uses guide RNAs consisting of a 5' protein scaffold followed by the target-specific guide, whereas Cas9 uses a 5' target-specific guide followed by a 3' protein-recognition aptamer. They rationalize that it should be possible to generate fusion guide RNAs that could work with both proteins by combining these features into one sgRNA, so called fusion guide (fg) RNAs. The fgRNAs should then be capable of functioning with both proteins. The indel frequency of the fgRNAs with the the Cas9 and Cpf1 proteins appear quite similar to those obtained with the single guide RNAs. The mismatch tolerance of the fgRNAs also are quite similar to the conventional guides of Cpf1 and Cas9. They then extend the target sequence site length to encompass to two target sites, one for Cpf1 and one for Cas9, thus expanding the use of this approach to enable independent targeting of 2 loci by both Cas9 and Cpf1. The only drawback with this design is that there is a decrease (~2-fold) in targeting efficiency in some cases when this modification is incorporated into the fgRNA, compared to the sgRNAs with the single protein. They then demonstrate orthogonality of the system by using an fgRNA to disrupt a target locus and also activate transcription from a second one. The data support well the conclusions.

One question I have for the authors is to define the nature of the fgRNA when bound to Cpf1 and Cas9. I suspect that the fgRNA gets trimmed and shortened and I think this information is important to know for the optimizing of dual targeting sites. This is not a difficult experiment to perform and would significantly add to the MS.

The paper needs to be carefully edited. For example, in the first sentence of the Abstract, "proteins" should read "protein". As well on line 39, "...truncated guide RNAs in the Cas9 protein can reduce the mismatch tolerance..." makes no sense – the authors probably meant "...truncated guide RNAs can reduce the mismatch tolerance...".

Reviewer #2:

Remarks to the Author:

The manuscript describes the demonstration that guide RNAs for Cpf1 and SpCas9 gene editing nucleases can be joined together and still retain their targeting functions. This is perhaps not surprising, since bases beyond the 3' end of Cpf1 crRNA were not expected to be important, and the 5' end of the SpCas9 sgRNA was similarly not expected to be important. However, this is the first time this union has been shown. The experiments were very straightforward for any practitioner in this field, and there are no concerns about the work technically. However, the manuscript fails to explain why the concatenation of these guide RNAs is important. The results seem more a matter of scientific curiosity than an advance that will have sustained impact in the field or significantly change current practice. The study therefore seems more appropriate for a specialized journal.

Point-by-point Responses to Comments

Reviewer #1 (Remarks to the Author):

In this MS, the authors take advantage of the fact that Cpf1 uses guide RNAs consisting of a 5' protein scaffold followed by the target-specific guide, whereas Cas9 uses a 5' target-specific guide followed by a 3' protein-recognition aptamer. They rationalize that it should be possible to generate fusion guide RNAs that could work with both proteins by combining these features into one sgRNA, so called fusion guide (fg) RNAs. The fgRNAs should then be capable of functioning with both proteins. The indel frequency of the fgRNAs with the the Cas9 and Cpf1 proteins appear quite similar to those obtained with the single guide RNAs. The mismatch tolerance of the fgRNAs also are quite similar to the conventional guides of Cpf1 and Cas9. They then extend the target sequence site length to encompass to two target sites, one for Cpf1 and one for Cas9, thus expanding the use of this approach to enable independent targeting of 2 loci by both Cas9 and Cpf1. The only drawback with this design is that there is a decrease (~2-fold) in targeting efficiency in some cases when this modification is incorporated into the fgRNA, compared to the sgRNAs with the single protein. They then demonstrate orthogonality of the system by using an fgRNA to disrupt a target locus and also activate transcription from a second one. The data support well the conclusions.

One question I have for the authors is to define the nature of the fgRNA when bound to Cpf1 and Cas9. I suspect that the fgRNA gets trimmed and shortened and I think this information is important to know for the optimizing of dual targeting sites. This is not a difficult experiment to perform and would significantly add to the MS.

Response: We agree that fgRNA would be trimmed or shortened by Cpf1 or Cas9 protein. Because Cpf1 protein has RNase activity, fgRNA might also be cleaved. To investigate whether fgRNA could be cleaved by LbCpf1 protein, we performed an *in vitro* cleavage assay. We did not detect any cleaved fgRNA fragment after spCas9 and/or Cpf1 protein treatment (Supplementary Fig. 2).

We have stated these results in result section on page 5 as follows:

“In contrast to Cas9, Cpf1 possesses both RNase and DNase activities, which allow it to process its own pre-crRNA and subsequently use the processed RNA to recognize and cleave the target DNA. We also confirmed that fgRNAs recruit both Cpf1 and Cas9 proteins without any modification and might function with both Cpf1 and Cas9 proteins at each targeted locus (Supplementary Fig. 2).”

The paper needs to be carefully edited. For example, in the first sentence of the Abstract, “proteins” should read “protein”. As well on line 39, “...truncated guide RNAs in the Cas9 protein can reduce the mismatch tolerance...” makes no sense – the authors probably meant “...truncated guide RNAs can reduce the mismatch tolerance...”.

Response: We have now revised the first sentence of the abstract and have carefully revised “proteins” or “protein” in the entire manuscript. Based on the suggestion, we have now deleted the phrase “in the Cas9 protein” on page 3, line 13.

Reviewer #2 (Remarks to the Author):

The manuscript describes the demonstration that guide RNAs for Cpf1 and SpCas9 gene editing nucleases can be joined together and still retain their targeting functions. This is

perhaps not surprising, since bases beyond the 3' end of Cpf1 crRNA were not expected to be important, and the 5' end of the SpCas9 sgRNA was similarly not expected to be important. However, this is the first time this union has been shown. The experiments were very straightforward for any practitioner in this field, and there are no concerns about the work technically. However, the manuscript fails to explain why the concatenation of these guide RNAs is important. The results seem more a matter of scientific curiosity than an advance that will have sustained impact in the field or significantly change current practice. The study therefore seems more appropriate for a specialized journal.

Response: We respect reviewer's opinion, but we think that fgRNAs have important implications in CRISPR fields. We have added the following paragraph in the Discussion section on page 6 to explain the importance of fgRNAs:

“CRISPR/Cas systems are widely used in various biological research fields for genome manipulation, and several efforts have been made to improve the nature of these systems. Cas9 and Cpf1 are representative proteins of the CRISPR system and have distinct characteristics. Because both Cas proteins have different guide RNA structures, it is necessary to make independent guide RNAs even if they recognize the same target sequences. In this study, we developed and validated fgRNAs—new guide RNA constructs that can work with both Cas9 and Cpf1 proteins. In addition, by extending the target sequences of fgRNAs, we demonstrated that fgRNAs can be used for targeting two independent endogenous sites using Cas9 and Cpf1 proteins.

Although Cas9 and Cpf1 proteins might be complementary because of their distinct characteristics, there are no attempt to use the orthogonality of these two Cas proteins, yet. We demonstrated that fgRNAs, which allow us to use the orthogonality of Cas9

and Cpf1 proteins with one guide RNA construct, could induce gene disruption and activation simultaneously and with this concept, a variety of applications will be possible. For example, if fgRNAs are designed to direct dCas9 variants to target key factors of DNA repair mechanisms, the fgRNAs may be used to alter the mutation frequency or pattern of Cpf1. Taken together, fgRNAs have the potential to simplify multiple gene manipulation with various Cas proteins and expand the use of genome editing tools in biological research.”

Reviewers' Comments:

Reviewer #1:

Remarks to the Author:

The authors have completely missed my point. I wasn't interested as to whether Cpf1 could cleave the fgRNA in an in vitro setting (totally artificial), I want them to define, following expression of Cpf1 and Cas9 and the fgRNA IN A CELL, what does the fgRNA look like that is protein-bound. For this, they will need to perform RNA immunoprecipitations using antibodies to Cpf and Cas9 (or tagged variants), followed by RNA seq.

Also, my comment regarding editing of the MS is not restricted to the few examples I cited, the entire MS needs to be carefully edited (which from their response does not seem to have happened).

Reviewer #2:

Remarks to the Author:

Unfortunately, the authors have offered little in their response to elevate the impact of this study. Cas9 nuclease is frequently used to make "large deletions", in which two gRNAs are used to excise an intervening DNA segment. There have been several reports of high-throughput libraries of paired gRNAs used in CRISPR screens. Catalytically dead Cas9 (dCas9) is quite often used with 4 - 6 gRNAs to perform synergistic gene activation. There is nothing unusual or limiting about using multiple gRNAs. Given the already ubiquitous use of multiple gRNAs, the joining of these two guide-RNAs does not seem to overcome a major obstacle or enable any fundamentally new capability. There was not one application in the authors' response that could not be done currently without fgRNAs, and the use of fgRNAs would provide only incremental simplicity to the procedure.

This assessment should not diminish the quality of the science nor the support of the conclusions by the data. However, the review policy states that "there should be a discernible reason why the work deserves the visibility of publication in a Nature journal rather than the best of the specialist journals." Many, many advances in CRISPR/Cas technology have been published in specialist journals, and the authors have not made the case for why this study should not be among them.

Point-by-point Responses to Comments

Reviewer #1 (Remarks to the Author):

The authors have completely missed my point. I wasn't interested as to whether Cpf1 could cleave the fgRNA in an *in vitro* setting (totally artificial), I want them to define, following expression of Cpf1 and Cas9 and the fgRNA IN A CELL, what does the fgRNA look like that is protein-bound. For this, they will need to perform RNA immunoprecipitations using antibodies to Cpf and Cas9 (or tagged variants), followed by RNA seq.

Response: We are grateful for reviewer's thoughtful comments. According to the previous reviewer's comments, we performed the *in vitro* experiments to know the lengths of fgRNAs which were bounded to Cpf1 or Cas9 proteins and might be trimmed. We think that the *in vitro* experiments are more unbiased methods because unknown factors may influence to the results *in vivo*. To our knowledge, most of papers about CRISPR/Cas system showed *in vitro* crRNA processing data, not *in vivo* data (E. Deltcheva et al. Nature 471, 602 (2011), L. Cong et al. Science 339, 819 (2013), I. Fonfara et al. Nature 532, 817 (2016), B. Zetsche et al. Nature Biotechnology 35, 31 (2017))

Of course, some unknown factors (i.e. host RNase III) IN A CELL may cleave or

process fgRNAs. But there were no study regarding to guide RNAs-processing host factors in mammalian cells. We think that it is beyond the scope of this study. Although here in our manuscript, we first report proof-of-concept for fgRNA, nature of fgRNA in living cells can be validated for application of fgRNAs in the future work

Also, my comment regarding editing of the MS is not restricted to the few examples I cited, the entire MS needs to be carefully edited (which from their response does not seem to have happened).

Response: We apologize for our insufficient response. We had also edited our entire manuscript through English language editing service for English language, grammar, punctuation, spelling, and tenses including the few examples in reviewer's comments. We will carefully discuss this issue with editor.

Thank you to reviewer for good comments to improve our manuscript.

Reviewer #2 (Remarks to the Author):

Unfortunately, the authors have offered little in their response to elevate the impact of this study. Cas9 nuclease is frequently used to make "large deletions", in which two gRNAs are used to excise an intervening DNA segment. There have been several reports of high-throughput libraries of paired gRNAs used in CRISPR screens. Catalytically dead Cas9 (dCas9) is quite often used with 4 - 6 gRNAs to perform synergistic gene activation. There

is nothing unusual or limiting about using multiple gRNAs. Given the already ubiquitous use of multiple gRNAs, the joining of these two guide-RNAs does not seem to overcome a major obstacle or enable any fundamentally new capability. There was not one application in the authors' response that could not be done currently without fgRNAs, and the use of fgRNAs would provide only incremental simplicity to the procedure.

Response: As reviewer's comments, there are many studies using multiple guide RNAs with Cas9 or Cas9 variants and we also showed that two pairs of guide RNAs could induce indels and reduce off-target effects with paired-Cas9 nickases (Genome Res. 2014 Jan; 24(1): 132–141). However, in this study, the key advance is the usage of the orthogonality of **two distinct types of Cas proteins using one guide RNA construct**.

Previous studies, as in reviewer's comment, showed that **single type** of Cas protein (i.e. Cas9 or Cpf1) could be used in multiplex genome manipulation with multiple gRNAs. In this study, however, using fgRNAs, not only multiple gene knockout but also combinations of gene knockout and activation could be successfully performed with two **distinct types** of Cas protein (Cas9+Cpf1 or dCas9 variants+Cpf1). We previously mentioned those advances in DISCUSSION section but, to emphasize more strongly, we have now revised DISCUSSION section on page 8 as below and all changes were highlighted.

“CRISPR/Cas systems are widely used in various biological research fields for genome manipulation, and several efforts have been made to improve the nature of these systems. For examples, Cas9 and Cpf1, which are representative proteins of the CRISPR system, are engineered for altering PAM sequences and improving target specificities. Also,

catalytically inactive Cas9 and Cpf1 are developed and widely used as programmable DNA binding proteins for transcriptional and epigenetic perturbations. As well as Cas proteins engineering, their guide RNAs are also improved to increase indel frequencies, target specificities, and induce multiple genome editing.

Cas9 and Cpf1 proteins have distinct characteristics, among which they have different guide RNA structures. For this reason, it is necessary to make independent guide RNAs even if they recognize the same target sequences. This is why there are no attempt to use the orthogonality of these two different types of Cas proteins, even though they might be complementary because of their distinct characteristics. In this study, we developed and validated fgRNAs—new guide RNA constructs that can work with both Cas9 and Cpf1 proteins. In addition, by extending the target sequences of fgRNAs, we demonstrated that fgRNAs can be used for targeting two independent endogenous sites using Cas9 and Cpf1 proteins. We also demonstrated that fgRNAs could be used to induce gene disruption and activation simultaneously. With this concept, we believe that a variety of applications will be possible. For example, if fgRNAs are designed to direct dCas9 variants to key factors of DNA repair mechanisms and Cpf1 to wanted-target genes, it may be possible to alter the mutation frequencies or patterns of Cpf1. In addition to the fgRNA itself, fgRNAs combined with the above mentioned engineered-Cas proteins and improved-guide RNAs technologies would give rise to various possibilities applications. Taken together, fgRNAs have the potential to simplify multiple gene manipulation with various Cas proteins and expand the use of genome editing tools in biological research.”

Thank you to reviewer for insightful comments to improve our manuscript.

Reviewers' Comments:

Reviewer #1 (Remarks to the Author)

It is important to assess if the fusion guide RNAs are trimmed in cells, not in vitro. It is in cells that this system will be used. The concern, and the reason for this request, is that a bi-functional guide RNA when bound to only one target will be trimmed in cells by nucleases and thus unable to participate in a second reaction involving the second target. This is known to occur in the case of spCas9. If you make sgRNAs that cover 30 nucleotides of target sequence, these get trimmed down to ~20 nucleotides (we have also performed this experiment and observed the same result). Hence, I do feel it is important to determine the proportion of fgRNAs in a cell that can potentially participate in both reactions...otherwise the authors are just making guide RNAs that bind to one protein or the other, and as a consequence of trimming only function in one reaction. This is not much better than co-expressing two different sgRNAs.

Point-by-point Responses to Comments

Reviewer #1 (Remarks to the Author):

It is important to assess if the fusion guide RNAs are trimmed in cells, not in vitro. It is in cells that this system will be used. The concern, and the reason for this request, is that a bi-functional guide RNA when bound to only one target will be trimmed in cells by nucleases and thus unable to participate in a second reaction involving the second target. This is known to occur in the case of spCas9. If you make sgRNAs that cover 30 nucleotides of target sequence, these get trimmed down to ~20 nucleotides (we have also performed this experiment and observed the same result). Hence, I do feel it is important to determine the proportion of fgRNAs in a cell that can potentially participate in both reactions...otherwise the authors are just making guide RNAs that bind to one protein or the other, and as a consequence of trimming only function in one reaction. This is not much better than co-expressing two different sgRNAs.

Response: We have already shown in vitro assays that the fgRNAs were not processed by LbCpf1 and/or spCas9, but we investigated published references to infer the nature of fgRNAs in a cell. Cong et al., in their previous paper, speculated that natural 5'-directed-repeat (DR) in the guide RNAs of spCas9 would have been processed by unknown endogenous factors in human cells. However, we could not find a published article that other 5'-artificial sequences, not the natural 5'-DR, of guide RNAs were processed in human cells. Nissim et al., On the other hand, showed a reporter assay result in which 5'-additional sequences were not processed in human cells. We agree with your insightful advice that the fgRNAs may be trimmed in cells by

unknown endogenous factors, but we do not think that LbCfp1 or spCas9 alone can process the fgRNAs, as demonstrated in our in vitro cleavage assays. For our next fgRNAs-based application study, it will be necessary to determine the nature of fgRNAs in cells and we will consider your concerns. We greatly appreciate your helpful comments and results.

Reference.

1. L. Cong et al. *Science* 339, 819 (2013)
2. L. Nissim et al. *Molecular Cell* 54,698 (2014)